# Determining the Competitiveness of Spa-Centers in Order to Achieve Sustainability Using a Fuzzy Multi-Criteria Decision-Making Model

**Miroslav Mijajlović** [1] , **Adis Puška** [2] , **Željko Stević** [3,*] , **Dragan Marinković** [4] ,
**Dragan Doljanica** [5] , **Saša Virijević Jovanović** [5] , **Ilija Stojanović** [6] and **Jasminka Beširović** [2]

1   Faculty of Mechanical Engineering, University of Niš, A. Medvedeva 14, 18000 Niš, Serbia;
    miroslav.mijajlovic@masfak.ni.ac.rs
2   Government of Brčko district B&H, Bulevara mira 1, 76100 Brčko District, Bosnia and Herzegovina;
    adispuska@yahoo.com (A.P.); jasnahba@yahoo.com (J.B.)
3   Faculty of Transport and Traffic Engineering, University of East Sarajevo, Vojvode Mišića 52, 74000 Doboj,
    Bosnia and Herzegovina
4   Faculty of Mechanical Engineering and Traffic Systems, TU Berlin, Str. d. 17. Juni 135, 10623 Berlin,
    Germany; dragan.marinkovic@tu-berlin.de
5   MEF Faculty of Applied Management, Economics and Finance, University Business Academy in Novi Sad,
    Jevrejska 24, 11000 Belgrade, Serbia; dragan.doljanica@mef.edu.rs (D.D.); sasa.virijevic@mef.edu.rs (S.V.J.)
6   College of Business Studies, Al Ghurair University, Dubai International Academic City PO Box 37374, UAE;
    ilija.stojanovic1976@gmail.com
*   Correspondence: zeljkostevic88@yahoo.com or zeljko.stevic@sf.ues.rs.ba

**Abstract:** Bosnia and Herzegovina (B&H) possess many natural resources that can be exploited for the development of medical tourism. The offer of medical tourism in B&H is focused on spa tourism. B&H has 16 registered spa-centers offering different types of services. This study provides a complete overview of the assessment of the current state of spa-centres using expert decision-making and methods of multi-criteria analysis. An innovative and novel MCDM model based on integration of the FUCOM (full consistency method) and fuzzy MARCOS (Measurement of Alternatives and Ranking according to COmpromise Solution) methods was used. The model consists of 16 alternatives and eight sustainable criteria. The results of this research have shown that the spa-centers of Ilidža near Sarajevo, Fojnica and Vrućica have the best assessments of the current situation and prerequisites for sustainable business. These spa-centers should be a benchmark to other spas providing direction on how to improve their business to be more sustainable and competitive in the market. These esults were confirmed by a sensitivity analysis with two approaches used. The first approach was to compare the results obtained by the fuzzy MARCOS method with other fuzzy methods, and the second approach was to examine the influence of the application of different weights on the final ranking of the spa. The results of this study can serve spa-canter managers to understand the position of their spa-centers in order to exploit advantages they have and eliminate the shortcomings to improve their business.

**Keywords:** spa-centers; Bosnia and Herzegovina; sustainable tourism; FUCOM method; fuzzy MARCOS method

## 1. Introduction

Contemporary trends in tourism require specific tourism products that emphasize authenticity, uniqueness and intact resources [1]. Bosnia and Herzegovina (B&H) owns significant resources that

are not used adequately and in a sustained way. Especially, B&H possesses many natural, thermal and thermo-mineral springs and peloids, among them are very rare mineral springs, which are known in professional circles around the world [2]. B&H has significant comparative advantages in the field of tourism where competitiveness can be improved. The offer of spas is a type of medical tourism that needs to be improved in order to improve tourism to achieve sustainability in B&H.

The use of thermo-mineral waters in B&H dates to the distant past. Since the ancient times of the Greeks and Romans, the healing properties of geothermal springs have been noted [2]. The spa offer in B&H is being built on these natural resources. The main activity of spa-centers is the health function of treatments, which is the most important and oldest function of spas. However, more attention is paid to the importance of the spa offer for the purpose of sustaining tourism development. The priority of spa development should be built on the health and tourism function available to spa-centers in B&H. Furthermore, the development of spa-centers is conducive with development of the local tourist offer [3].

Although B&H has a rich tradition in medical tourism, in previous studies, a complete overview of the spas on offer in B&H has not been given. This paper seeks to evaluate the entire offer of medical tourism in B&H through the offering of spas. In this way, we will provide an insight into the current state of medical tourism and what B&H currently has in terms of medical tourism. Obtaining this information is necessary for the implementation of future activities to improve medical tourism in B&H by improving the spa offer. In addition, this paper will indicate the possibilities for improving medical tourism in B&H.

The aim of this paper is to develop guidelines for improving the spa offers in B&H in order to improve the competitiveness of spas. Improvements in competitiveness will be achieved by attracting more tourists and patients and young people to spas. In order to improve the spas on offer in B&H, it is necessary to evaluate the current and potential spas on offer. The evaluation of the spas on offer in B&H was performed by using an innovative multi-criteria decision-making model. The goal of this model is not to determine which is the best spa-center in B&H, but to determine the advantages and disadvantages that spas have. The results could form a solid basis to improve the competitiveness of spas in B&H.

This paper aims to address the following questions: (a) What is the current situation of the spa sector in B&H? (b) What are the advantages and disadvantages of spas? (c) What are the fundamentals on which to build a competitive advantage in spa-centres? The expected scientific contribution of this study is to evaluate the current situation and provide guidelines for the development of the spas on offer in B&H. Based on these guidelines, every spa manager can develop a business plan to improve competitiveness of their spa-center on the market. In addition, they will gain insights into the position of their spa-center relative to other spas. This study will assist in developing medical tourism in order to build competitive spa offerings in B&H. By strengthening the competitiveness of medical tourism, it will strengthen the overall tourism in B&H.

In order to achieve the study goals, the evaluation of selected spa-centers was performed by using a fuzzy approach. The fuzzy approach was adapted to human thinking because grading is done by applying linguistic values. Selected spa-centers, through a multi-criteria evaluation model, were evaluated by the expert. This evaluation model was used to evaluate spas in B&H. Additionally, a contribution of this paper is the development of one integrated Full COnsistency Method (FUCOM)-Fuzzy MARCOS (Measurement of Alternatives and Ranking according to COmpromise Solution) model.

## 2. A Theoretical Framework

The offer of Spas is a representative form of medical tourism [4], where various spa treatments are provided that include alternative therapies such as: homeopathy, osteopathy, acupuncture, yoga, counselling, fitness, aromatherapy, beauty treatments, aesthetic treatments, cosmetic surgery, liposuction, and chiropractic treatments [5]. Spa tourism is a narrower term than health tourism and

implies a type of health tourism that is carried out in spa centers in order to treat certain diseases, improve psycho-physical health or relaxation of the body [6]. Spas are used not only by those seeking a cure for diseases such as arthritis, back pain, obesity, trauma, asthma, sterility, and surgical rehabilitation, but also by those seeking relaxation, beauty and longevity treatments [7].

New health concepts and wider activities that emerged from the trends of modern Western societies, have contributed to the growth of spas and the development of the specialization and segmentation of spas [8]. According to the International Spa Association (ISPA), spas are classified into seven main categories [9]:

- Daily spas: include facial and body treatment services where overnight stays are not provided.
- Hotel spas, spa resorts: offer fitness and health services and spa services with the possibility of overnight stay.
- Spa destinations: the main goal of these spas is to direct visitors to an individual healthy lifestyle. This service can be achieved through a comprehensive program that includes spa services, fitness activities, health education, healthy cuisine and programs of special interest to users.
- Medical spas: are health centers in which professional on-site medical workers provide comprehensive medical and health services that integrate all types of therapies and treatments.
- Spa clubs: are facilities that have the purpose of fitness and offer professional spa services that are offered daily. It should be noted that hotels, gyms and fitness clubs are not spas, unless they explicitly offer spa products and services as an added offer.
- Mineral spas: offer natural mineral, thermal or sea water used in hydrotherapy treatments.
- Spas on cruise ships: provide professional spa services, fitness and health components and a selection of spa menus on cruise ships.

Development of the spa is considered a natural response to human desire for treatments in the context of the evolution of consciousness, globalization, and various global crises [10]. The diversity of spas has been influenced by the fact that spas are used not only by people seeking treatment for various diseases, but also by guests who want relaxation, beauty and longevity treatments [7]. On this basis, spa tourism has developed as one of the sectors of health tourism. Spa tourism is one of the oldest types of tourism that has been constantly evolving. Complex and various forms have developed under the influence of political and economic systems, on the one hand, and changes in social options and tourist interests in relation to this type of tourism, on the other [11].

Health tourism is a comprehensive concept for the application of medical and treatment tourism in which the main motivation of tourists is to improve and maintain their health [12]. However, there is no single definition of what spa tourism is [13]. Spa tourism can be defined as a part of health tourism that refers to the provision of specific services that include mineral and thermal waters, but is also used for leisure, because it offers accommodation services [7]. Jahić and Selimović [6] point out that spa tourism is a narrower term than health tourism and implies a type of health tourism that is carried out in spa centers in order to treat certain diseases, improve psycho-physical health or relax the body. As a result, more and more people visit spa centers to improve their health. On this basis, spa tourism is currently one of the fastest growing sub sectors in health tourism [14]. In many Western European countries, spa tourism is an important factor in local and regional development [13].

B&H is known for its sources of mineral water, which has been used in the treatment of various types of diseases [2]. Spa centers in B&H with a rich history have been built near these springs. There are currently 16 registered medical spas in B&H. These spas are the basis for development of medical tourism in B&H. Therefore, it is necessary to actively invest in the modernization of spa facilities in order to be able to offer guests quality spa services. It is necessary to improve medical treatments in spas and expand the range of treatments in order to improve the competitiveness of spas in B&H. In previous studies, only certain spas in B&H were included and no comparison was made between them. In previous scientific publications, medical tourism has not been made the focus of

attention and this paper represents a shift at improving this form of tourism. In the last ten years, only a few papers on medical tourism in B&H have been published, and some of them are presented below.

In his paper, Segić [15] provided a more precise definition of spa and recreational health tourism on a scientific and professional basis for the purpose of developing health tourism in the Republic of Srpska through foreign investments. Zelenbabić [16] took Spa Vrućica as an example of how the combination of natural resources and financial investments in the long run will produce positive results for: service providers, users and the local community. In their paper, Operta and Hyseni [17] presented the problems encountered with spa tourism in B&H, as well as the possibilities for improving tourist destinations. Bodiroža and Ćerketa [18] pointed out that the further development of health, spa and climate tourism should be based on modern world achievements, but above all in the construction of modern spa centers.

Spahić and Temimović [19] pointed out that tourism development in B&H should be based on the offer of spas which represents the backbone of the tourism development strategy. Jahić and Selimović [6] focused on the balenology and health tourism in Fojnica, which is based on thermal water, springs resources. In their paper, they provided guidelines on how to improve the competitiveness of the spa in Fojnica by offering modern treatments in the form of expanding health treatments to sports and recreational tourism. It is necessary to look at the current offer of spas in B&H, and they provided recommendations on how to improve and make this branch of tourism more competitive. Milinković et al. [20] based on the example of Spa Vrućica, tried to answer whether this spa-center can meet the modern needs of the tourist market, and thus contribute to economic growth and development of the national and local economy. Puška et al. [2] presented the spa offer in B&H in their paper, but they did not compare individual spas, rather they found some areas in which great opportunities are present for improving the spa offer in B&H.

Based on this literature review, it can be concluded that B&H has a rich range of spas on offer that can be used for the development of health tourism, but it is necessary to improve and enhance this offer of spas. Therefore, it is necessary to look at the current state of the spas on offer in B&H and provide guidelines on how to develop individual spa centers in order to be more competitive in the market, thus contributing to further development of health tourism in B&H. For this purpose, a multi-criteria approach for evaluation of the spa offer in B&H was used.

## 3. Methodology

The phases and methodology used in this paper consists of the following phases (Figure 1):

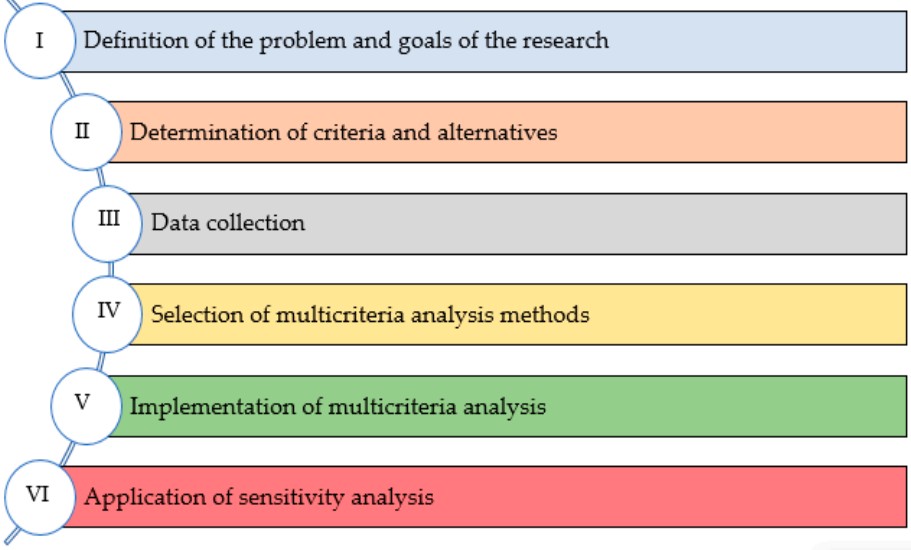

**Figure 1.** Phases of the research.

The initial phase of any study is to define the problem and goals of the research. The problem of this research is the evaluation of the current spa offer as a part of the health tourism industry in B&H. The goal of this paper is to develop guidelines for improving the spas on offer in B&H in order to improve the competitiveness of spa-centers. In order to evaluate the existing spa offer, it is necessary to define the criteria for evaluation of the current spa offer. When using this approach, a multi-criteria analysis of research alternatives is used. All registered spa-centers in B&H were taken as alternatives. The criteria used in this analysis were taken in collaboration with experts, as follows:

- Accommodation capacity (C1) is the number of rooms and beds that the spa offers
- Food and beverage offer (C2) represent various food and beverage services offered to guests of these spa-centers
- Internet promotion (C3) includes the existence of its own website and presence on specialized websites
- The offer of treatment and therapy (C4) includes the existence of different treatments and the availability of therapies in these spas
- Cosmetic treatments (C5) includes a comprehensive range of cosmetic and beauty and rejuvenation treatments
- Recreation and relaxation (C6) include the offer of the spa in a form of various services aimed at the relaxation of spa users
- Education and events (C7) include the possibility of conducting various seminars, congresses and various entertainment events in spas
- Natural conditions (C8) includes the comprehensiveness of natural resources available to individual spas.

In order to collect the data necessary for evaluation of the current spa offer in B&H, expert evaluation was used. Experts were selected from the pool of researchers who worked on the project: "Development and promotion of health spa tourism in the cross-border area of Bosnia and Herzegovina–Serbia." A total of three experts in the field of health tourism were selected. A two-part questionnaire was sent to these experts. The first part was related to the weight of the criteria of this model, while the second part of the questionnaire was related to the current state of the spa according to the presented criteria. These estimates are presented in the form of linguistic value (Table 1). Fuzzy logic was used to transform these linguistic values. The use of fuzzy logic was done with the fuzzy MARCOS method to rank the spas according to expert evaluation. The values of the criteria weights were determined using the FUCOM model.

**Table 1.** Linguistic values and affiliation function to fuzzy number [21].

| Linguistic Values | Fuzzy Numbers |
| --- | --- |
| Very bad (VB) | (0,0,1) |
| Bed (B) | (0,1,3) |
| Medium bed (MB) | (1,3,5) |
| Medium (M) | (3,5,7) |
| Medium good (MG) | (5,7,9) |
| Good (G) | (7,9,10) |
| Very good (VG) | (9,10,10) |

After the data were collected from the experts, the methods from multi-criteria analysis were implemented and the research results were obtained. Sensitivity analysis was performed to confirm the results. Sensitivity analysis in this paper had the following tasks: to examine the sensitivity of the rank order to changing the weights of the criteria and to examine the sensitivity of the data on application of different methods of multicriteria analysis. More details on the steps of the above methods of multicriteria analysis are shown below.

### 3.1. FUCOM (Full Consistency Method) Method

The FUCOM method is a new model for determining the weight of criteria in multicriteria decision making. The FUCOM method uses the comparison of paired criteria and the validation of results by deviating from the maximum consistency, developed by Pamučar et al. [22]. Using this method, the subjectivity in the decision-making process is reduced. This method, in relation to other methods for determining the subjective weights of criteria, has the following main advantages: reduction in the number of pairs for comparison, consistency in comparing criteria and contribution to rational judgment [23,24]. The FUCOM method is implemented using the following steps [25,26]:

Step 1. Ranking of criteria/sub-criteria using expert judgment.

Step 2. Determining the vector of comparative significance of the evaluation criteria.

Step 3. Defining the constraints of a nonlinear optimization model. The values of the weighting coefficients should satisfy two conditions, namely:

- Condition 1. The ratio of the weight coefficients is equal to the comparative significance between the observed, that the condition is fulfilled: $w_k/w_{k+1} = \varphi_{k/(k+1)}$
- Condition 2. The final values of the weighted coefficients should satisfy the condition of mathematical transitivity $\varphi_{k/(k+1)} \times \varphi_{(k+1)/(k+2)} = [yellow]\varphi_{k/(k+2)}$

Step 4. Defining a model for determining the final values of the weighting coefficients of the evaluation criteria.

Step 5. Solving the model and obtaining the final weight of the criteria/sub-criteria $(w_1, w_2, \ldots, w_n)^T$.

### 3.2. Fuzzy MARCOS (Measurement of Alternatives and Ranking According to Compromise Solution) Method

The MARCOS method was developed by Stević et al. [27] and it represents a new method of multicriteria decision making. The MARCOS method is based on a defined relationship between alternatives and reference values that are presented as ideal and anti-ideal points. Decision making using the MARCOS method is based on the utility function [28]. The utility function looks at an alternative to an ideal and anti-ideal solution. The best alternative is the one that is closest to the ideal and at the same time the furthest from the anti-ideal solution.

The steps for calculating the fuzzy MARCOS method are as follows [29]:

Step 1. Forming an initial fuzzy decision matrix.

Step 2. Expanding the initial fuzzy decision matrix. In this step, the initial matrix is expanded with anti-ideal (AAI) and ideal solution (AI). The anti-ideal solution (AAI) is an alternative with the worst characteristic depending on the type of criteria. The ideal solution (AI) is an alternative with the best characteristic.

The anti-ideal solution (AAI) is obtained by applying the following expression:

$$\widetilde{A}(AI) = \min_i \widetilde{x}_{ij} \; if \; j \in \; B \; and \; \max_i \widetilde{x}_{ij} \; if \; j \in C \tag{1}$$

while the ideal solution (AI) is obtained using the following expression:

$$\widetilde{A}(ID) = \max_i \widetilde{x}_{ij} \; if \; j \in \; B \; and \; \min_i \widetilde{x}_{ij} \; if \; j \in C \tag{2}$$

*B* represents the criteria that need to be maximized, while *C* represents the criteria that need to be minimized.

Step 3. Normalizing the initial fuzzy decision matrix. Normalization is performed using the following expressions:

$$\widetilde{n}_{ij} = \left(n_{ij}^l, n_{ij}^m, n_{ij}^u\right) = \left(\frac{x_{id}^l}{x_{ij}^u}, \frac{x_{id}^l}{x_{ij}^m}, \frac{x_{id}^l}{x_{ij}^l}\right) if \; j \in \; C \tag{3}$$

$$\widetilde{n}_{ij} = \left(n_{ij}^l, n_{ij}^m, n_{ij}^u\right) = \left(\frac{x_{ij}^l}{x_{id}^u}, \frac{x_{ij}^m}{x_{id}^u}, \frac{x_{ij}^u}{x_{id}^u}\right) \ if \ j \in B \tag{4}$$

where $l$ is the first fuzzy number, $m$ is the second fuzzy number and $u$ is the third fuzzy number.

Step 4. Weighting the normalized decision matrix. The weighting is done using the following expression:

$$\widetilde{v}_{ij} = \left(v_{ij}^l, v_{ij}^m, v_{ij}^u\right) = \widetilde{n}_{ij} \otimes \widetilde{w}_j = \left(n_{ij}^l \times w_j^l, n_{ij}^m \times w_j^m, n_{ij}^u \times w_j^u\right) \tag{5}$$

Step 5. Calculation of the $S_i$ matrix which implies summation of values by rows or alternatives including alternatives for anti-ideal and ideal solution by the following expression:

$$\widetilde{S}_i = \sum_{i=1}^n \widetilde{v}_{ij} \tag{6}$$

Step 6. Calculation of the degree of usefulness $K_i$ according to the anti-ideal and ideal solution using the following expressions:

$$\widetilde{K}_i^- = \frac{\widetilde{S}_i}{\widetilde{S}_{ai}} = \left(\frac{s_i^l}{s_{ai}^u}, \frac{s_i^m}{s_{ai}^m}, \frac{s_i^u}{s_{ai}^l}\right) \tag{7}$$

$$\widetilde{K}_i^+ = \frac{\widetilde{S}_i}{\widetilde{S}_{id}} = \left(\frac{s_i^l}{s_{id}^u}, \frac{s_i^m}{s_{id}^m}, \frac{s_i^u}{s_{id}^l}\right) \tag{8}$$

Step 7. Calculation of the fuzzy matrix $\widetilde{T}_i$. using the following expression:

$$\widetilde{T}_i = \widetilde{t}_i = \left(t_i^l, t_i^m, t_i^u\right) = \widetilde{K}_i^- \oplus \widetilde{K}_i^+ = \left(k_i^{-l} + k_i^{+l}, k_i^{-m} + k_i^{+m}, k_i^{-u} + k_i^{+u}\right) \tag{9}$$

Determining the fuzzy number $\widetilde{D}$ using the following expression:

$$\widetilde{D} = \left(d^l, d^m, d^u\right) = \max_i \widetilde{t}_{ij} \tag{10}$$

Step 8. De-fuzzification of fuzzy numbers using the following expression:

$$df_{crisp} = \frac{l + 4m + u}{6} \tag{11}$$

Step 9. Determining the utility function $f\left(\widetilde{K}_i\right)$ through the aggregation of utility functions according to the anti-ideal solution (a) and the ideal solution (b).

- Utility function according to the anti-ideal solution

$$f\left(\widetilde{K}_i^+\right) = \frac{\widetilde{K}_i^-}{df_{crisp}} = \left(\frac{k_i^{-l}}{df_{crisp}}, \frac{k_i^{-m}}{df_{crisp}}, \frac{k_i^{-u}}{df_{crisp}}\right) \tag{12}$$

- Utility function according to the ideal solution

$$f\left(\widetilde{K}_i^-\right) = \frac{\widetilde{K}_i^+}{df_{crisp}} = \left(\frac{k_i^{+l}}{df_{crisp}}, \frac{k_i^{+m}}{df_{crisp}}, \frac{k_i^{+u}}{df_{crisp}}\right) \tag{13}$$

Step 10. Calculation of the final utility function:

$$f(K_i) = \frac{K_i^+ + K_i^-}{1 + \frac{1 - f\left(K_i^+\right)}{f\left(K_i^+\right)} + \frac{1 - f\left(K_i^-\right)}{f\left(K_i^-\right)}}; \tag{14}$$

Step 11. Ranking alternatives. The best alternative is the one with the highest value, while the worst is the alternative with the lowest value.

## 4. Results

When assessing the current condition of spas in B&H, it was first necessary to determine the weights of the main criteria and we used the FUCOM method in this study.

### 4.1. Criteria Weights Obtained Using FUCOM Method

According to the steps from FUCOM method, the experts first had to determine which, in their opinion, is the most important criterion and this criterion was assigned a value of one. The other criteria were assigned values in relation to the importance towards the most important criterion. Experts had at their disposal, values from one to nine, with the possibility to assign decimal values to the criteria. The less important the criterion, its closer its designated value to nine. It should be noted that it was possible that the criteria have the same importance and in such cases they were assigned the same value. Based on these steps, the experts evaluated the criteria (Table 2).

**Table 2.** Evaluation of criteria by experts.

| Expert 1 | C4 | C8 | C2 | C6 | C5 | C7 | C1 | C3 |
| --- | --- | --- | --- | --- | --- | --- | --- | --- |
| | 1 | 1 | 2 | 2 | 3 | 3 | 4 | 4 |
| Expert 2 | C1 | C4 | C8 | C6 | C2 | C5 | C3 | C7 |
| | 1 | 1.1 | 1.1 | 1.4 | 1.6 | 1.6 | 2 | 2.5 |
| Expert 3 | C4 | C8 | C7 | C6 | C5 | C1 | C2 | C3 |
| | 1 | 1 | 5 | 6 | 7 | 8 | 8 | 8 |

As can be seen from the value of the criteria, there is no agreement between the experts on are the most important criteria for assessing the current state of spas in B&H. This can be noticed from the example in which expert one and expert three have indicated the criterion C4 (treatment offer and therapy) as the most important criterion, while expert two has indicated the criterion C1 (accommodation capacity). In order to harmonize these opinions, individual values of weights were calculated for the criteria by individual experts, and these values were harmonized using a geometric mean. The procedure for implementing the FUCOM method has been previously explained, and in this part of the paper only the final weights by criteria are presented.

The most important criterion according to the expert is C4 (treatment offer and therapy), followed by criterion C8 (natural conditions), and the least important criterion according to them is criterion C6 (recreation and relaxation) (Table 3).

**Table 3.** Criteria weight values.

| | C1 | C2 | C3 | C4 | C5 | C6 | C7 | C8 |
| --- | --- | --- | --- | --- | --- | --- | --- | --- |
| **Expert 1** | 0.1078 | 0.1274 | 0.1078 | 0.1557 | 0.1168 | 0.1274 | 0.1168 | 0.1402 |
| **Expert 2** | 0.1545 | 0.1312 | 0.1264 | 0.1377 | 0.1312 | 0.0605 | 0.1209 | 0.1377 |
| **Expert 3** | 0.1000 | 0.1000 | 0.1000 | 0.1889 | 0.1063 | 0.1133 | 0.1214 | 0.1700 |
| **Joint Grade** | 0.1199 | 0.1200 | 0.1121 | 0.1612 | 0.1189 | 0.0966 | 0.1210 | 0.1502 |

### 4.2. Ranking Alternatives Using Fuzzy MARCOS Method

After determining a weight of the criteria, the experts had to assess the current state of the spas on offer in B&H. Experts evaluated each spa-center and assigned an appropriate linguistic value (Table 1).

The experts obtained the necessary information for evaluation of these spas by searching the internet, visiting these spas and using similar methods. Based on this knowledge, the experts evaluated these spas and the current state of the spa sector in B&H (Table 4). In order to obtain the results of the assessment, it was necessary to transform the linguistic values into fuzzy numbers using the affiliation functions to fuzzy numbers (Table 1).

**Table 4.** Linguistic assessment of the current state of spas in B&H by experts.

| Expert 1 | C1 | C2 | C3 | C4 | C5 | C6 | C7 | C8 |
|---|---|---|---|---|---|---|---|---|
| A1—Spa Dvorovi | G | G | MG | MG | MG | G | MG | G |
| A2—Spa Fojnica | VG | G | VG | VG | MG | MG | VG | VG |
| A3—Spa Gata | MG | G | MG | G | G | G | MB | G |
| A4—Spa Guber | VB | B | VB | VB | VB | VB | VB | VG |
| A5—Spa Ilidža–Gradačac | MG | MG | G | G | MG | MG | G | G |
| A6—Spa Akvaterm–Olovo | MG | MG | G | G | G | G | MG | MG |
| A7—Spa Ilidža–Sanski Most | MG | MG | MG | MG | MB | G | MB | VG |
| A8—Spa Ilidža near Sarajevo | VG | VG | VG | G | VG | G | VG | VG |
| A9—Spa Kulaši | G | G | G | MG | MG | G | MB | MG |
| A10—Spa Laktaši | MG | G | G | G | MG | G | MB | G |
| A11—Spa Lješljani | MB | G | VB | B | VB | MB | VB | MG |
| A12—Spa Mlječanica | G | MB | B | MB | B | G | G | G |
| A13—Spa Slatina | VG | G | MB | VG | MB | MB | MG | G |
| A14—Spa Vilina Vlas | MG | MG | MG | MG | MB | MG | MG | G |
| A15—Spa Vrućica | VG | G | G | G | MG | G | VG | VG |
| A16—Slana Spa | MG | G | MG | G | MG | G | G | MG |
| **Expert 2** | **C1** | **C2** | **C3** | **C4** | **C5** | **C6** | **C7** | **C8** |
| A1—Spa Dvorovi | G | G | G | MG | G | MG | G | MG |
| A2—Spa Fojnica | VG | G | VG | G | VG | G | VG | G |
| A3—Spa Gata | MG | MG | MB | MG | G | G | G | G |
| A4—Spa Guber | VB | B | B | VB | VB | VB | VB | VG |
| A5—Spa Ilidža–Gradačac | G | G | VG | G | G | G | G | G |
| A6—Spa Akvaterm–Olovo | MG | MG | MG | G | MG | MG | MG | G |
| A7—Spa Ilidža–Sanski Most | MG | G | MG | MG | G | G | MB | G |
| A8—Spa Ilidža near Sarajevo | VG | G | VG | G | VG | G | VG | G |
| A9—Spa Kulaši | G | G | G | MG | G | G | G | G |
| A10—Spa Laktaši | G | G | G | MG | G | MG | G | MG |
| A11—Spa Lješljani | MB | MG | B | B | B | MB | VB | G |
| A12—Spa Mlječanica | MG | MG | MB | G | G | G | MG | G |
| A13—Spa Slatina | VG | G | MB | VG | MG | MG | MG | G |
| A14—Spa Vilina Vlas | MG | G | MG | MG | MG | G | MG | VG |
| A15—Spa Vrućica | VG | G | G | G | VG | G | G | G |
| A16—Slana Spa | G | G | MG | G | VG | G | G | MG |
| **Expert 3** | **C1** | **C2** | **C3** | **C4** | **C5** | **C6** | **C7** | **C8** |
| A1—Spa Dvorovi | G | MG | MG | MG | MG | MG | G | MG |
| A2—Spa Fojnica | VG | G | VG | VG | G | G | VG | VG |
| A3—Spa Gata | MG | MG | MG | G | G | MG | G | G |
| A4—Spa Guber | B | B | MB | G | G | G | MG | VG |
| A5—Spa Ilidža–Gradačac | MG | G | G | G | MG | MG | MG | MG |
| A6—Spa Akvaterm–Olovo | G | G | G | MG | MG | MG | MG | G |
| A7—Spa Ilidža–Sanski Most | MG | MG | MG | MG | G | G | MB | MB |
| A8—Spa Ilidža near Sarajevo | VG | VG | VG | VG | G | G | VG | VG |
| A9—Spa Kulaši | MG | G | MG | G | MG | G | G | G |
| A10—Spa Laktaši | G | MG | G | MG | G | G | G | MG |
| A11—Spa Lješljani | B | MB | B | B | B | MB | VB | G |
| A12—Spa Mlječanica | MB | G | MB | MG | G | MB | G | MG |
| A13—Spa Slatina | VG | MG | MB | G | G | MG | G | G |
| A14—Spa Vilina Vlas | G | G | MG | G | MG | G | MG | G |
| A15—Spa Vrućica | G | VG | G | VG | G | G | G | G |
| A16—Slana Spa | MG | G | G | VG | G | G | G | MG |

Since certain spas received the values of Very Bad (VB) and Poor (P), it was not possible to use a geometric mean to match these scores, but it an arithmetic mean was applied, and a common fuzzy decision matrix was formed (Table 5). Once this matrix was formed, the next step was to expand this decision matrix by calculating the ideal and anti-ideal solution. Following this, the next step was to normalize the data. Since the score for each of the used criteria should have been as high as possible, these criteria were normalized using expression (3). The next step was to make the normalized decision matrix (expression 5).

**Table 5.** Initial fuzzy decision matrix.

|      | C1           | C2           | C3           | C4           | C5           | C6           | C7           | C8           |
| ---- | ------------ | ------------ | ------------ | ------------ | ------------ | ------------ | ------------ | ------------ |
| A1   | 4.3 6.3 8.0  | 6.3 8.3 9.7  | 5.7 7.7 9.3  | 5.0 7.0 9.0  | 5.7 7.7 9.3  | 5.7 7.7 9.3  | 6.3 8.3 9.7  | 5.7 7.7 9.3  |
| A2   | 9.0 10.0 10.0| 7.0 9.0 10.0 | 9.0 10.0 10.0| 8.3 9.7 10.0 | 7.0 8.7 9.7  | 6.3 8.3 9.7  | 9.0 10.0 10.0| 8.3 9.7 10.0 |
| A3   | 5.0 7.0 9.0  | 5.7 7.7 9.3  | 3.7 5.7 7.7  | 5.0 7.0 8.7  | 3.0 5.0 7.0  | 3.7 5.7 7.7  | 2.3 4.3 6.3  | 4.3 6.3 8.0  |
| A4   | 0.0 0.3 1.7  | 0.0 1.0 3.0  | 0.3 1.3 3.0  | 1.0 1.7 3.0  | 1.0 1.7 3.0  | 1.0 1.7 3.0  | 1.7 2.3 3.7  | 9.0 10.0 10.0|
| A5   | 5.7 7.7 9.3  | 6.3 8.3 9.7  | 7.7 9.3 10.0 | 7.0 9.0 10.0 | 5.7 7.7 9.3  | 5.7 7.7 9.3  | 5.0 7.0 8.7  | 6.3 8.3 9.7  |
| A6   | 5.7 7.7 9.3  | 5.7 7.7 9.3  | 6.3 8.3 9.7  | 6.3 8.3 9.7  | 4.3 6.3 8.3  | 5.7 7.7 9.3  | 5.0 7.0 9.0  | 3.7 5.7 7.7  |
| A7   | 5.0 7.0 9.0  | 5.7 7.7 9.3  | 5.0 7.0 9.0  | 5.0 7.0 9.0  | 2.3 4.3 6.3  | 4.3 6.3 8.0  | 1.0 3.0 5.0  | 5.7 7.3 8.3  |
| A8   | 9.0 10.0 10.0| 8.3 9.7 10.0 | 9.0 10.0 10.0| 7.7 9.3 10.0 | 8.3 9.7 10.0 | 7.0 9.0 10.0 | 9.0 10.0 10.0| 8.3 9.7 10.0 |
| A9   | 6.3 8.3 9.7  | 7.0 9.0 10.0 | 6.3 8.3 9.7  | 5.7 7.7 9.3  | 5.7 7.7 9.3  | 7.0 9.0 10.0 | 5.0 7.0 8.3  | 6.3 8.3 9.7  |
| A10  | 6.3 8.3 9.7  | 6.3 8.3 9.7  | 7.0 9.0 10.0 | 4.3 6.3 8.3  | 5.0 7.0 8.7  | 5.0 7.0 8.7  | 2.3 4.3 6.3  | 4.3 6.3 8.3  |
| A11  | 0.7 2.3 4.3  | 3.0 5.0 7.0  | 0.0 0.7 2.3  | 0.0 1.0 3.0  | 0.0 0.7 2.3  | 1.0 3.0 5.0  | 0.0 0.0 1.0  | 6.3 8.3 9.7  |
| A12  | 4.3 6.3 8.0  | 3.0 5.0 7.0  | 0.7 2.3 4.3  | 4.3 6.3 8.0  | 2.0 3.7 5.7  | 3.7 5.7 7.3  | 5.0 7.0 8.7  | 6.3 8.3 9.7  |
| A13  | 9.0 10.0 10.0| 6.3 8.3 9.7  | 1.0 3.0 5.0  | 8.3 9.7 10.0 | 3.0 5.0 7.0  | 3.7 5.7 7.7  | 5.7 7.7 9.3  | 7.0 9.0 10.0 |
| A14  | 5.7 7.7 9.3  | 6.3 8.3 9.7  | 5.0 7.0 9.0  | 4.3 6.3 8.3  | 3.7 5.7 7.7  | 5.0 7.0 8.7  | 5.0 7.0 9.0  | 7.7 9.3 10.0 |
| A15  | 8.3 9.7 10.0 | 7.7 9.3 10.0 | 7.0 9.0 10.0 | 7.7 9.3 10.0 | 7.0 8.7 9.7  | 7.0 9.0 10.0 | 7.7 9.3 10.0 | 7.7 9.3 10.0 |
| A16  | 5.7 7.7 9.3  | 7.0 9.0 10.0 | 5.7 7.7 9.3  | 7.7 9.3 10.0 | 7.0 8.7 9.7  | 7.0 9.0 10.0 | 7.0 9.0 10.0 | 5.0 7.0 9.0  |

What distinguishes the fuzzy MARCOS method from other similar methods is the calculation of the utility and the utility function. To implement this, it was necessary to sum up the values by alternatives for individual fuzzy numbers. The calculation of the degree of usefulness was done using expressions (7) and (8) (Table 6). The next step is calculating the fuzzy matrix $\widetilde{T}_i$ (Table 6) (expression 9). The utility values of the anti-ideal and ideal solution were summed for the alternatives and the maximum values of the individual fuzzy numbers were determined. After this, these maximum values of fuzzy numbers (expression 11) were de-fuzzified and the values were calculated $df_{\text{crisp}}$.

**Table 6.** Summarizing and calculating the degree of utility and Fuzzy matrix $\widetilde{T}_i$.

|            | $S_i$           | $K_i^-$          | $K_i^+$          |     | $\widetilde{T}_i$ |
| ---------- | --------------- | ---------------- | ---------------- | --- | ----------------- |
| Ideal      | 0.85 0.98 1.00  | 0.85 1.00 1.17   | 2.72 6.57 15.44  |     |                   |
| A1         | 0.56 0.76 0.92  | 0.56 0.77 1.08   | 1.77 5.08 14.21  |     | 2.32 5.86 15.29   |
| A2         | 0.81 0.95 0.99  | 0.81 0.97 1.16   | 2.56 6.36 15.33  |     | 3.37 7.32 16.50   |
| A3         | 0.41 0.61 0.80  | 0.41 0.63 0.94   | 1.32 4.13 12.35  |     | 1.73 4.76 13.29   |
| A4         | 0.20 0.27 0.40  | 0.20 0.28 0.47   | 0.63 1.83 6.14   |     | 0.82 2.11 6.60    |
| A5         | 0.62 0.82 0.95  | 0.62 0.84 1.12   | 1.97 5.49 14.71  |     | 2.59 6.33 15.83   |
| A6         | 0.53 0.73 0.90  | 0.53 0.75 1.06   | 1.69 4.91 13.92  |     | 2.22 5.66 14.98   |
| A7         | 0.43 0.63 0.80  | 0.43 0.64 0.94   | 1.37 4.21 12.42  |     | 1.80 4.85 13.36   |
| A8         | 0.83 0.97 1.00  | 0.83 0.99 1.17   | 2.65 6.50 15.44  |     | 3.48 7.49 16.61   |
| A9         | 0.61 0.81 0.95  | 0.61 0.83 1.11   | 1.95 5.46 14.65  |     | 2.56 6.30 15.76   |
| A10        | 0.50 0.70 0.87  | 0.50 0.72 1.02   | 1.59 4.71 13.39  |     | 2.10 5.43 14.40   |
| A11        | 0.15 0.27 0.44  | 0.15 0.28 0.52   | 0.47 1.84 6.85   |     | 0.62 2.12 7.37    |
| A12        | 0.38 0.57 0.75  | 0.38 0.59 0.87   | 1.21 3.85 11.52  |     | 1.59 4.43 12.39   |
| A13        | 0.57 0.75 0.87  | 0.57 0.77 1.02   | 1.83 5.05 13.49  |     | 2.40 5.82 14.52   |
| A14        | 0.54 0.73 0.90  | 0.54 0.75 1.05   | 1.71 4.92 13.86  |     | 2.25 5.67 14.91   |
| A15        | 0.75 0.92 1.00  | 0.75 0.94 1.17   | 2.39 6.20 15.38  |     | 3.15 7.15 16.55   |
| A16        | 0.65 0.84 0.97  | 0.65 0.86 1.13   | 2.07 5.65 14.91  |     | 2.72 6.51 16.04   |
| Anti-ideal | 0.06 0.15 0.31  | 0.06 0.15 0.37   | 0.21 1.00 4.86   | Max | 3.48 7.49 16.61   |

Calculating the value $df_{crisp}$ was done as follows: $df_{crisp} = (3.48 + 7.49 + 16.61)/6 = 8.34$. This value was used to calculate the utility function according to anti-ideal and ideal solutions. This was followed by calculating the value of the utility function. Calculating expressions $f(\widetilde{K}_i^-)$ was done by taking the values of the degree of usefulness according to the anti-ideal solution $(K_i^-)$ and dividing with the value $df_{crisp}$. The value of the expression $f(\widetilde{K}_i^+)$ was calculated by dividing the utility values according to the ideal solution $(K_i^+)$ with the value $df_{crisp}$ (Table 7).

**Table 7.** Utility functions.

|     | $f(\widetilde{K}_i^-)$ | $f(\widetilde{K}_i^+)$ |
|-----|------------------------|------------------------|
| A1  | 0.21 0.61 1.70 | 0.07 0.09 0.13 |
| A2  | 0.31 0.76 1.84 | 0.10 0.12 0.14 |
| A3  | 0.16 0.49 1.48 | 0.05 0.08 0.11 |
| A4  | 0.08 0.22 0.74 | 0.02 0.03 0.06 |
| A5  | 0.24 0.66 1.76 | 0.07 0.10 0.13 |
| A6  | 0.20 0.59 1.67 | 0.06 0.09 0.13 |
| A7  | 0.16 0.50 1.49 | 0.05 0.08 0.11 |
| A8  | 0.32 0.78 1.85 | 0.10 0.12 0.14 |
| A9  | 0.23 0.66 1.76 | 0.07 0.10 0.13 |
| A10 | 0.19 0.57 1.61 | 0.06 0.09 0.12 |
| A11 | 0.06 0.22 0.82 | 0.02 0.03 0.06 |
| A12 | 0.14 0.46 1.38 | 0.05 0.07 0.10 |
| A13 | 0.22 0.61 1.62 | 0.07 0.09 0.12 |
| A14 | 0.20 0.59 1.66 | 0.06 0.09 0.13 |
| A15 | 0.29 0.74 1.84 | 0.09 0.11 0.14 |
| A16 | 0.25 0.68 1.79 | 0.08 0.10 0.14 |

After all the necessary parameters were calculated, the fuzzy numbers were de-fuzzified, namely the values of the degree of utility (DSK$^-$ and DSK$^+$), the functions of the degree of utility (Df(K$^-$) and Df(K$^+$)) and the value of the final utility function were calculated (K$_i$) (Table 8).

**Table 8.** Results of the fuzzy MARCOS method.

|     | $DSK^-$ | $DSK^+$ | $Df(\widetilde{K}_i^-)$ | $Df(\widetilde{K}_i^+)$ | $K_i$ | Rank |
|-----|---------|---------|-------------------------|-------------------------|-------|------|
| A1  | 0.788 | 6.051 | 0.725 | 0.094 | 0.6238 | 7  |
| A2  | 0.973 | 7.220 | 0.866 | 0.117 | 0.9391 | 2  |
| A3  | 0.644 | 5.030 | 0.603 | 0.077 | 0.4170 | 13 |
| A4  | 0.296 | 2.347 | 0.281 | 0.035 | 0.0860 | 16 |
| A5  | 0.847 | 6.441 | 0.772 | 0.102 | 0.7182 | 5  |
| A6  | 0.763 | 5.877 | 0.705 | 0.091 | 0.5851 | 10 |
| A7  | 0.656 | 5.106 | 0.612 | 0.079 | 0.4319 | 12 |
| A8  | 0.994 | 7.347 | 0.881 | 0.119 | 0.9779 | 1  |
| A9  | 0.842 | 6.409 | 0.768 | 0.101 | 0.7104 | 6  |
| A10 | 0.731 | 5.640 | 0.676 | 0.088 | 0.5362 | 11 |
| A11 | 0.298 | 2.448 | 0.293 | 0.036 | 0.0904 | 15 |
| A12 | 0.600 | 4.687 | 0.562 | 0.072 | 0.3598 | 14 |
| A13 | 0.779 | 5.921 | 0.710 | 0.093 | 0.6028 | 8  |
| A14 | 0.764 | 5.877 | 0.705 | 0.092 | 0.5861 | 9  |
| A15 | 0.949 | 7.097 | 0.851 | 0.114 | 0.8978 | 3  |
| A16 | 0.870 | 6.596 | 0.791 | 0.104 | 0.7580 | 4  |

According to the results obtained by applying the integrated model of the FUCOM method and fuzzy MARCOS, we have found that the best ratings have been assigned to the alternative A8 (Spa Ilidža near Sarajevo), followed by the alternative A2 (Spa Fojnica), while spa A4 (Spa Guber) has rated with the worst position.

*4.3. Sensitivity Analysis*

Sensitivity analysis was performed in two ways. The first way was focused to compare the results obtained using the fuzzy MARCOS method with the other five fuzzy methods, namely: simple additive weighting (SAW) [30], weighted aggregated sum product assessment (WASPAS) [30], multi-attributive border approximation area comparison (MABAC) [26], additive ratio assessment (ARAS) [30] and technique for order preference by similarity ideal solution (TOPSIS) [31]. Another way was focused to change the weights of the criteria and the impact of these weights on the ranking order of the alternatives [32].

The first way of conducting sensitivity analysis has aimed to show the sensitivity of the ranking order based on the use of different methods of multi-criteria analysis. The results of this method of conducting sensitivity analysis examined the obtained results of the primary method used in the analysis. The results of this sensitivity analysis have shown that the ranking order of the first seven alternatives have been the same by applying any fuzzy method (Figure 2). The correlation between the ranking orders of alternatives using different fuzzy methods was examined using the Spearman correlation coefficient (SCC) [33] (Table 9).

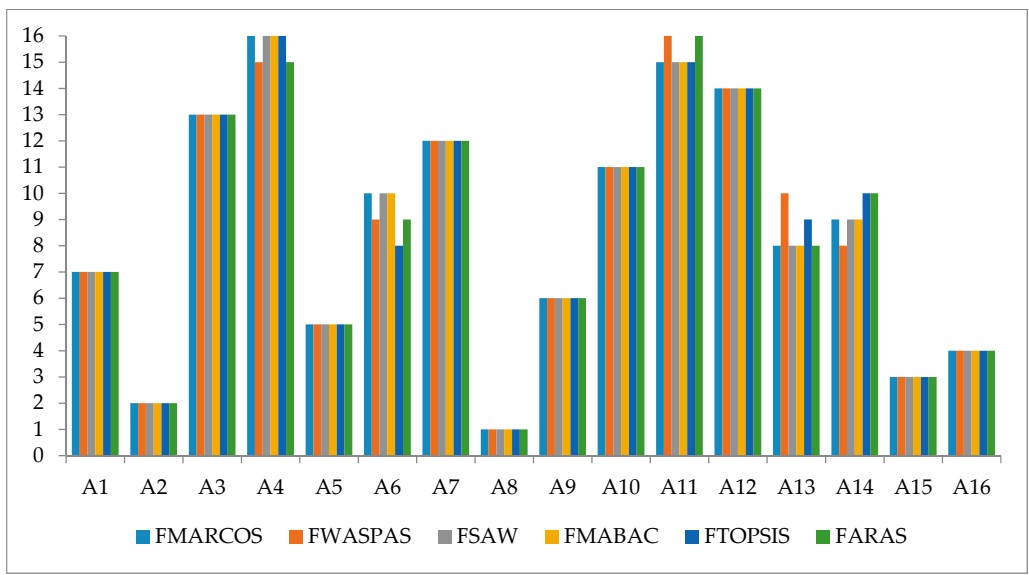

**Figure 2.** Results of sensitivity analysis using different fuzzy methods.

**Table 9.** Spearman correlation coefficient results.

|  | FMARCOS | FWASPAS | FSAW | FMABAC | FTOPSIS | FARAS | Average |
|---|---|---|---|---|---|---|---|
| FMARCOS | 1.000 | 0.988 | 1.000 | 1.000 | 0.991 | 0.994 | 0.995 |
| FWASPAS |  | 1.000 | 0.988 | 0.988 | 0.988 | 0.988 | 0.990 |
| FSAW |  |  | 1.000 | 1.000 | 0.991 | 0.994 | 0.996 |
| FMABAC |  |  |  | 1.000 | 0.991 | 0.994 | 0.995 |
| FTOPSIS |  |  |  |  | 1.000 | 0.994 | 0.997 |
| FARAS |  |  |  |  |  | 1.000 | 1.000 |

Another way to conduct sensitivity analysis is to change the weight of the criteria and observe the effects of change on rankings of the alternatives. This sensitivity analysis eliminates the subjective attitude of experts regarding the importance of the weights of individual criteria. The implementation of this method of sensitivity analysis was implemented as follows. First, scenarios were formed with the assumption: one criterion is three times more important than the other criteria and thus eight scenarios are formed. The ninth scenario assumes that all criteria are of equal importance (Table 10).

**Table 10.** Scenarios in conducting sensitivity analysis.

|  | C1 | C2 | C3 | C4 | C5 | C6 | C7 | C8 |
|---|---|---|---|---|---|---|---|---|
| Scenario 1 | 0.300 | 0.100 | 0.100 | 0.100 | 0.100 | 0.100 | 0.100 | 0.100 |
| Scenario 2 | 0.100 | 0.300 | 0.100 | 0.100 | 0.100 | 0.100 | 0.100 | 0.100 |
| Scenario 3 | 0.100 | 0.100 | 0.300 | 0.100 | 0.100 | 0.100 | 0.100 | 0.100 |
| Scenario 4 | 0.100 | 0.100 | 0.100 | 0.300 | 0.100 | 0.100 | 0.100 | 0.100 |
| Scenario 5 | 0.100 | 0.100 | 0.100 | 0.100 | 0.300 | 0.100 | 0.100 | 0.100 |
| Scenario 6 | 0.100 | 0.100 | 0.100 | 0.100 | 0.100 | 0.300 | 0.100 | 0.100 |
| Scenario 7 | 0.100 | 0.100 | 0.100 | 0.100 | 0.100 | 0.100 | 0.300 | 0.100 |
| Scenario 8 | 0.100 | 0.100 | 0.100 | 0.100 | 0.100 | 0.100 | 0.100 | 0.300 |
| Scenario 9 | 0.125 | 0.125 | 0.125 | 0.125 | 0.125 | 0.125 | 0.125 | 0.125 |

Applying the scenario setting, a sensitivity analysis was performed (Figure 3).

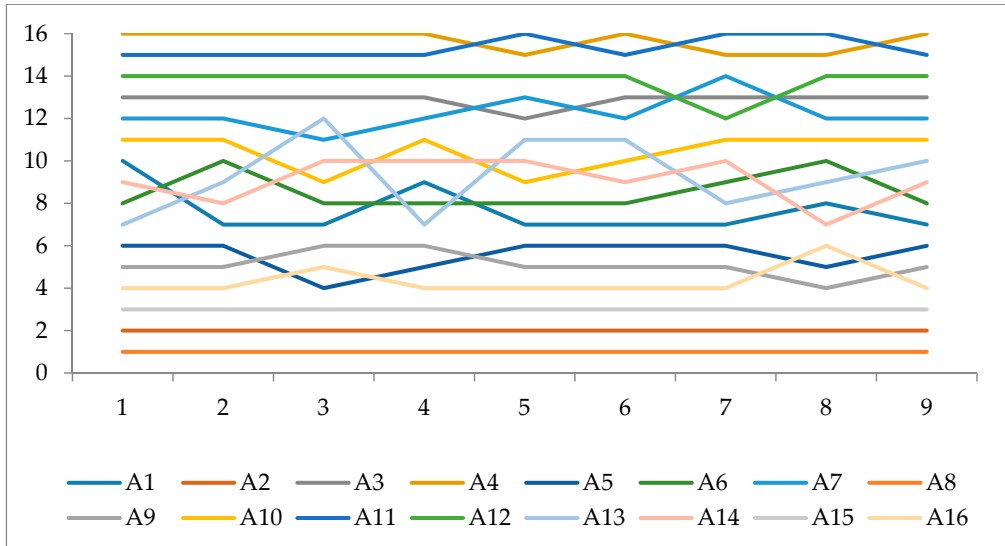

**Figure 3.** Results of sensitivity analysis for set scenarios.

Managers of all spa-centers can use the study results to understand the current state of their spa compared to the position of other spa-centers in B&H. Based on this insight, they are able to perceive their advantages and disadvantages, take competitive advantages and eliminate the shortcomings.

## 5. Discussion

Based on the obtained results using FUCOM-Fuzzy MARCOS model, it can be noticed that three spas are especially distant from the others: Ilidža near Sarajevo, Fojnica and Vrućica. This is because these spas have used natural resources and have incorporated these resources within the offer of their services. This is the reason why these spa-centers differ from other spas. The situation is the same at the bottom of the ranking list, where two spas especially deviate from other spas: Guber and Spa Lješljani. These spas have good natural resources, but accommodation capacities are inadequate or not exist, and the supply is weak or non-existent. These are the main reasons why such research results have been obtained.

After conducting the first phase of sensitivity analysis, it can be concluded that only five alternatives underwent a change in ranking order using different methods. The fuzzy WASPAS method has deviated the most in the ranking order of five alternatives, the fuzzy ARAS method in the ranking order of four alternatives and the fuzzy TOPSIS method in ranking order of three alternatives. Other fuzzy methods MARCOS, SAW and MABAC have provided the same ranking order for all alternatives.

The results of the Spearman correlation coefficient have shown that the fuzzy method MARCOS, MABAC and SAW deviates the most with the fuzzy WASPAS method (r = 0.988), followed by the fuzzy TOPSIS method (r = 0.991). One of the characteristics of the fuzzy WASPAS method is not having a perfect connection with other methods because the ranking order obtained by this method deviates from other methods, and this is also present in the fuzzy TOPSIS and ARAS methods. Thus, the results obtained using the fuzzy MABAC method and the ranking of spa-centers in the analysis were confirmed.

The results of the first method of sensitivity analysis have shown that there is a small difference between the used fuzzy methods (average r = 0.996), thus the results of the study were confirmed. Alternatives A6, A13 and A14 have provided the largest deviation in the ranking. We have found that alternatives A6, A13 and A14 have similar characteristics in terms of assessing the current situation; therefore, this difference in ranking orders has occurred by individual fuzzy methods. We found a similar example with alternatives A4 and A11 which have been ranked as the worst alternatives.

The results of the second part of the sensitivity analysis have shown that alternatives A8, A2 and A15 are not sensitive to changes in criterion weights, and these spas maintained the same ranking order. This finding has shown that the assessment of the current condition of these spas is the best in relation to other spas and that they are insensitive to changes in the weight of the criteria. These spas have also shown the best characteristics, and in order to further improve their competitiveness, it is necessary to improve those criteria where they did not receive maximum ratings. As an example, Ilidža Spa near Sarajevo should improve the offer of treatments and therapies (C4) and cosmetic treatments (C5); however this spa should also improve the criterion of natural conditions (C8), but this cannot be improved because one spa cannot change the environment where it is located and all the predispositions offered by that environment.

The second group of spas are spa-centers A5, A9 and A16, in which there were changes in the rankings in relation to individual scenarios. Thus, the spa A16 has been placed in fourth place in seven scenarios, while in scenarios three and eight it has been placed in fourth place. These results have shown that the spa A9 has better natural conditions than the spa A16, while the spa A5 has better internet promotion. In order to improve its competitiveness, the A16 spa should primarily work on internet promotion and be more accessible to all potential and future users as well as current users of the services of this spa. The spa A9 has shown the best ratings in natural conditions and this is the advantage from which this spa can benefit in order to improve competitiveness. Similarly, all evaluated spa-centers can use the results of this study to work on improving their business. It should be noted that this sensitivity analysis has shown that all 16 spas can be grouped into five groups. In order to move to a better group, the spa must invest in its development, and only in that way it can be more competitive and achieve better business results.

The disadvantages of this analysis can be found in a restricted number of experts involved and only certain MCDM methods used in this study. However, the aim of this paper was not to include as many experts as possible, to assess the current state of the spa offer, in order to improve the competitiveness of medical tourism in B&H. Moreover, the involvement of a larger number of experts potentially leads to greater inconsistencies in the opinions and possibilities of obtaining even conflicting opinions of experts. Therefore, the opinions of experts who worked together on the project "Development and promotion of health spa tourism in the cross-border area of Bosnia and Herzegovina–Serbia" were taken. Through the completion of this research, they are informed about the current situation regarding the spa offers in B&H. Therefore, the opinion of these experts, who are deeply involved in this area, is fully competent to provide assessments of the current state of medical tourism in B&H.

The use of different methods showed that the results obtained using the fuzzy MARCOS method are similar to the results obtained by other methods, and there are no significant deviations in the results. In this way, these results were confirmed, which served to provide guidelines for improving the competitiveness of medical tourism in B&H. Based on the findings from this study, it is necessary to implement certain measures in order to improve the spa offers in B&H, and thus improve medical

tourism in B&H, especially due to the fact that spas offer the highest volume of services in medical tourism in B&H.

## 6. Conclusions

In order to improve their business, all business entities must first assess the current situation in relation to the competition to understand their own advantages and disadvantages. This paper provides a complete overview of the current state of all registered spa-centers in B&H. An innovative decision model based on the integration of the FUCOM and fuzzy MARCOS methods was implemented. The results have shown that the spa-centers of Ilidža near Sarajevo, Fojnica and Vrućica have the best assessments of the current situation. These spa-centers should be an example to other spas on how to use their resources to improve their business. The conducted sensitivity analysis confirmed the results of our research and provided guidelines on which criteria need to be improved by individual spas.

In addition to assessment of the current situation, a new method for multi-criteria analysis was used in this paper, namely fuzzy MARCOS. With sensitivity analysis we compared the results of this method with the results given by other fuzzy methods. It has been found that these results did not differ significantly. Thus, it has been proven that the fuzzy MARCOS method can be used in solving problems that require multi-criteria decision making. This paper also has shown that different methods can be integrated into a single model that will make decision-making easier for decision makers.

The limitation of this research is that the analysis was based on expert opinion. This decision-making is subjective due to the potential tendency that experts sometimes aspire to give better grades to some alternatives. Therefore, in future decisions it is necessary to include the opinions of users of these spas. However, this research is the first to give a complete picture of the current state of spas in B&H. Previous studies have analyzed only a partial overview of individual spas; thus, this paper makes a significant contribution to improving the business of spas in B&H. Another shortcoming of this paper are the criteria used. However, these criteria were chosen by experts in order to reduce the volume of used criteria, but keeping relevant numbers of those adequate for getting appropriate results to assess the current state of the spa sector in B&H.

In future research, it is necessary to include more criteria in order to get the best possible grade and to provide additional information to spa-centers' managers. The proposed current situation assessment model has shown significant flexibility and could be used in future research where multi-criteria decision-making needs to be applied. Furthermore, additional studies in the field of the spa-sector of B&H should be implemented in order to get specific guidelines on how to use the natural resources in B&H in strengthening medical tourism with the final aim to improve this branch of the economy.

**Author Contributions:** Conceptualization, A.P. and J.B.; methodology, A.P. and Ž.S.; validation, D.M. and I.S.; formal analysis, D.M.; investigation, A.P. and J.B.; writing—original draft preparation, A.P., M.M. and D.D.; writing—review and editing, Ž.S. and S.V.J.; supervision, D.D. and M.M. All authors have read and agreed to the published version of the manuscript.

**Funding:** This research received no external funding.

**Conflicts of Interest:** The authors declare no conflict of interest.

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
