# Peer review of "Determining the Competitiveness of Spa-Centers in Order to Achieve Sustainability Using a Fuzzy Multi-Criteria Decision-Making Model"

_sustainability, doi:10.3390/su12208584_

Round 1
Reviewer 1 Report
Dear authors:
Thnks for your submision.
I found that the Introduction does not contain all the necessary information. As scientific standards, the Introduction must answer the questions: What was I studying? Why was it an important question? What did we know about it before I did this study? And, How will this study advance our knowledge? Try to focus on a wider perspective and use your case study as an example. Introduction should have no more than one page, so the reader will be able to read it quickly.
A theoretical framework should be included with part of the information presented in the introduction, observing relevant scientific contributions from the last 10 years with similar experiences.
The methodology is well described but I believe that the results of the Sensitivity Analysis should be included in the results section so that the discussion section offers a vision of the problem and the necessary nuances that must be carried out as well as the limitations that this type of analysis provokes.
The methodology is well described but I believe that the results of the Sensitivity Analysis should be included in the results section so that the discussion section offers a vision of the problem and the necessary nuances that must be carried out as well as the limitations that this type of analysis provokes.
Author Response
Reviewer 1:
Thank you very much for the useful suggestions. We accepted all of the suggestions and we are sure that this will improve the quality and contribute to a better understanding of the paper.
Dear authors:
Thanks for your submision.
Comment 1: I found that the Introduction does not contain all the necessary information. As scientific standards, the Introduction must answer the questions: What was I studying? Why was it an important question? What did we know about it before I did this study? And, How will this study advance our knowledge? Try to focus on a wider perspective and use your case study as an example. Introduction should have no more than one page, so the reader will be able to read it quickly.
Response to comment 1: Thank you for your suggestion. The introduction section is now the length of one page. We have tried to answer all questions. The introduction with newly added information now looks:
Contemporary trends in tourism require specific tourism products that emphasize authenticity, uniqueness and intact resources [1]. Bosnia and Herzegovina (B&H) own significant resources that are not used adequately and in a sustained way. Especially, B&H possesses many natural, thermal and thermo-mineral springs and peloids, among them are very rare mineral springs, which are known in professional circles around the world [2]. B&H has significant comparative advantages in the field of tourism where competitiveness can be improved. Spa offer is a type of medical tourism that needs to be improved in order to improve tourism in order to achieve sustainability in B&H.
The use of thermo - mineral waters in B&H dates to the distant past. Since the ancient times of the Greeks and Romans, the healing properties of geothermal springs have been noticed [2]. The spa offer in B&H is being built on these natural resources. The main activity of spa-centres is the health function of treatment, which is the most important and oldest function of spas. However, more attention is paid to the importance of the spa offer for the purpose of sustaining tourism development. The priority of spa development should be built on the health and tourism function available to spa-centres in B&H. Furthermore, the development of spa-centres is conducive with development of the local tourist offer [3].
Although B&H has a rich tradition in medical tourism, in previous studies it has not been given a complete overview of the spa offer in B&H. This paper seeks to evaluate the entire offer of medical tourism in B&H through the spa offer. In this way, we will provide an insight into the current state of medical tourism and what B&H currently has in terms of medical tourism. Obtaining this information is necessary for the implementation of future activities to improve medical tourism in B&H by improving the spa offer. In addition, this paper will indicate the possibilities for improving medical tourism in B&H.
The aim of this paper is to develop guidelines for improving the spa offer in B&H in order to improve the competitiveness of spas. Improving competitiveness will be achieved by attracting more tourists and patients and young people to spas. In order to improve the spa offer in B&H, it is necessary to evaluate the current and potential spa offer. The evaluation of the spa offer in B&H was performed by using an innovative multi-criteria decision-making model. The goal of this model is not to determine which is the best spa-centre in B&H, but to determine the advantages and disadvantages that spas have. The results could be a solid basis to improve the competitiveness of spas in B&H.
This paper aims to address the following questions: a) What is the current situation in the spa sector in B&H? b) What are the advantages and disadvantages of spas? c) What are the fundamentals on which to build a competitive advantage in spa-centres? The expected scientific contribution of this study is to evaluate the current situation and provide guidelines for the development of the spa offer in B&H. Based on these guidelines, every spa manager can develop a business to improve competitiveness of their spa-canter on the market. In addition, they will gain insight into the position of their spa-canter relative to other spas. This study will assist in developing medical tourism in order to build competitive spa offerings in B&H. By strengthening the competitiveness of medical tourism, it will strengthen the overall tourism in B&H.
In order to achieve the study goals, the evaluation of selected spa-centres was performed by using a fuzzy approach. The fuzzy approach was adapted to human thinking because grading is done by applying linguistic values. Selected spa-centres, through a multi-criteria evaluation model were evaluated by the expert. This evaluation model was used to evaluate spas in B&H. Also, as contribution of this paper can be represented development one integrated FUCOM-Fuzzy MARCOS model.
Comment 2: A theoretical framework should be included with part of the information presented in the introduction, observing relevant scientific contributions from the last 10 years with similar experiences.
Response to comment 2: Thank you for this suggestion. We have formed new section: 2. A theoretical framework. Please see pages 2-4, lines 87-171.
Comment 3: The methodology is well described but I believe that the results of the Sensitivity Analysis should be included in the results section so that the discussion section offers a vision of the problem and the necessary nuances that must be carried out as well as the limitations that this type of analysis provokes.
Response to comment 3: All sensitivity analysis has been included in section results. Please see subsection 4.3. Also, we have created new section: 5. Discussion. In this section, we have show discussion about obtained results and add new information about the practical implications and disadvantaged of the study. Please see pages 14 and 15.
Reviewer 2 Report
The presented research is very interesting. The study is solid.
Author Response
Reviewer 2:
Comment: The presented research is very interesting. The study is solid.
Response: Thank you for your positive review.
Reviewer 3 Report
The presentation of the case is suitable. Methodology is solid and presented in detail. The presentation of results and their analysis is good.
There are however some grammar errors that should be corrected before publication (mostly incorrect spelling, some questionable term selections).
Line 245:
"Step 4. Aggravating the normalized decision matrix. The aggravation is done using the 246 following expression:"
I am not familiar with the use of word "aggravation" in that context. Is it the proper technical term for the procedure?
Author Response
Reviewer 3:
Thank you very much for the useful suggestions. We accepted all of the suggestions and we are sure that this will improve the quality and contribute to a better understanding of the paper.
Comment 1: The presentation of the case is suitable. Methodology is solid and presented in detail. The presentation of results and their analysis is good.
Response to comment 1: Thank you for your positive comment.
Comment 2: There are however some grammar errors that should be corrected before publication (mostly incorrect spelling, some questionable term selections).
Line 245:
"Step 4. Aggravating the normalized decision matrix. The aggravation is done using the following expression:"
I am not familiar with the use of word "aggravation" in that context. Is it the proper technical term for the procedure?
Response to comment 2: We have polished the whole paper. Also, we have changed word "aggravation" in weighting. Please see line 260.
Round 2
Reviewer 1 Report
Well, thanks for consider my point of view. Now I think your paper provodes better comprenhension and background for the readers.
Best regardas.